# Effects of Mn-Depleted Zone Formation on Acicular Ferrite Transformation in Weld Metals under High Heat Input Welding

**DOI:** 10.3390/ma15238477

**Published:** 2022-11-28

**Authors:** Fengyu Song, Chaochao Yin, Fusheng Hu, Kaiming Wu

**Affiliations:** 1College of Physics, Mechanical and Electrical Engineering, Longyan University, Longyan 364012, China; 2The State Key Laboratory of Refractories and Metallurgy, Hubei Province Key Laboratory of Systems Science on Metallurgical Processing, International Research Institute for Steel Technology, Collaborative Center on Advanced Steels, Wuhan University of Science and Technology, Wuhan 430081, China

**Keywords:** weld metal, Mn-depleted zone, acicular ferrite, cooling rate

## Abstract

In this present work, during high heat input welding of the weld metal, different types of Mn-depleted zones were achieved by different cooling rates. The effects of cooling rates on Mn-depleted zone formation and acicular ferrite (AF) transformation were analyzed. The Mn-depleted zone around the inclusions, as well as the interface concentration of Mn atoms, are found to be significantly different with different cooling rates. When the cooling rate is 10 °C/s, the interface concentration of Mn atoms around the inclusions is the lowest, the area of Mn-depleted zone is the smallest, and the proportion of AF in the weld metal is the highest. As the cooling rate decreases further, the interface concentration of Mn begins to rise, the area of the Mn-depleted zone gradually expands, and the proportion of AF decreases. However, when the cooling rate reaches 100 °C/s, only a very small amount of MnS precipitates, no Mn-depleted zone forms around the inclusions, and acicular ferrite cannot be produced effectively in the weld metal.

## 1. Introduction

Welding of thick plate for large structures frequently uses a single-pass high heat input welding, such as automatic gas-electric vertical welding and electroslag welding [1,2]. The efficiency of this single-pass welding is nearly ten times that of traditional multi-pass welding. However, as the welding heat input increases, the peak temperature rises, and the cooling rate decreases, resulting in obvious coarsening of the microstructure of the weld metal and serious deterioration of mechanical properties, especially impact toughness [3]. Small non-metallic inclusions with high melting points are typically introduced during the steelmaking process to pin grain boundaries during heating and induce the formation of acicular ferrite during the solidification and cooling process. Acicular ferrite (AF) induced by inclusions plays a key role in microstructure refinement and low-temperature impact toughness improvement. Efficient welding and short manufacturing cycles for heavy plates can be achieved by introducing acicular ferrite under high heat inputs of 100~400 kJ/cm [4,5].

Researchers discovered that specific inclusions such as TiN, Ti_2_O_3_, Al_2_O_3_, MnS, V (CN), and RE-M (O, S) [6,7,8,9] can provide nucleation sites for the formation of AF [10,11,12,13,14]. The mechanism of AF formation induced by different types of inclusions is not the same. The common nucleation mechanisms of AF include a. solute-poor zone formation around inclusions, b. low interfacial energy mechanism and c. strain induction mechanism, and d. low lattice mismatch mechanism. Generally, mechanisms (a) and (d) are more acceptable by researchers.

Our previous studies have shown that (Ti-Al)O_x_-MnS promotes the nucleation of AF due to the Mn-depleted zone (MDZ) around the inclusion [15]. Researchers also obtained a similar conclusion for Al_2_O_3_–MgO–ZrO_2_–MnS and TiO*x*-MnS-Al_2_O_3_-CaO composite inclusions, where they attributed the formation of acicular ferrite to the depletion of Mn elements around the inclusions, which increased the nucleation driving force of acicular ferrite [16,17]. However, researchers mainly focus on the Mn-Depleted zone as the mechanism of acicular ferrite nucleation, but do not quantify the size of the Mn-Depleted zone and the quantitative relationship with the acicular ferrite. In this present work, the concentration distribution of Mn around inclusions was characterized by transmission electron microscopy with energy dispersive spectroscopy and theoretical calculations. The influences of cooling rates on MDZ formation and AF nucleation in the weld metal were discussed.

## 2. Experimental Procedure

The weld metal obtained by single gas-electric vertical welding under a high heat input of about 205 kJ/cm was used as the research object, and a high heat input flux-cored wire prepared in the laboratory was used in the welding process. Conventional Q235 steel was selected as base metal in this work. The specific chemical composition is C 0.16 %, Mn 0.45 %, Si 0.26 %, S 0.035%, and P 0.035%. Table 1 presents the composition of the weld metal used in this study. Four samples of φ3 × 10 mm size were cut from the center of the weld metal and then treated on a fully automatic Formastor-FII transformation temperature measuring apparatus. The samples were heated to 1400 °C and then held for 10 seconds to simulate the high temperature residence time of large heat input welding. Cooling from 1400 °C to 800 °C (austenitizing temperature) at different cooling rates (1 °C/s, 5 °C/s, 10 °C/s, and 100 °C/s) results in different amounts of precipitated MnS, that is to obtain different types of Mn-Depleted zone. The acicular ferrite is considered to be formed at about 600 °C. In the cooling process below 800 °C in this work, the cooling rate is determined as 2 °C/s because the author’s previous studies have shown that the weld metal is more likely to induce acicular ferrite when cooled at 2 °C/s. [15]. The diagram of heat treatment process is shown in Figure 1.

The heat-treated samples were cut by wire cutting, fixed by phenolic resin, and then ground and polished by sandpaper. A 3% (volume fraction) nital (4 mL of HNO_3_ + 96 mL of C_2_H_5_OH) was used to etch the samples. The metallographic microstructures of the samples near the thermocouple were detected by a LEICA Q5501W Microscope (Leica LTD, Heidelberg, Germany), and the proportions of AF in the samples were statistically measured. An FEI-SEM (Apreo S HiVac, FEI, Hillsboro, OR, USA) equipped with an EDS (AZteclive Ultim Max 100, OXFORD Instruments, Abingdon, UK) and a field-emission electron probe microscope (EPMA-1720H, Shimadzu Corporation, Kyoto, Japan) were used for inclusion characterizations. Thin film samples with a thickness of approximately 300 µm were obtained by wire cutting, and then 1500# sandpaper was used to grind the samples to a thickness of approximately 50 µm. A Struers TenuPol-5 electrolytic double spray thinning instrument was used to thin wafer samples of Φ3 mm diameter. The electrolytic double spray solution was a mixture of 9% perchloric acid and ethanol. A TEM (JEOL JSM-2000EX, JEOL Ltd, Tokyo, Japan) equipped with an energy-dispersive X-ray spectroscope (JEOL JEM-2100plus, JEOL Ltd, Tokyo, Japan) was employed to characterize inclusions and AF. The concentration distribution of Mn around inclusions and the area of the Mn-depleted zone were measured by EDS. In order to improve the measurement accuracy, multiple scans were taken at the same distance from the inclusion, and the average value was taken as the Mn concentration at the distance. A high-temperature confocal laser scanning microscope (HT-CLSM, Yonekura MFG. Co. Ltd., Osaka, Japan) was used to capture the phase transformation process in the weld metal. Before counting the proportion of microstructures, a specific tissue was marked according to the difference in microstructure morphology, and then its area fraction was counted with Photoshop software.

## 3. Results

### 3.1. Microstructure

Figure 2 displays the microstructure of the weld metal after continuous cooling. In this work, the size of acicular ferrite is usually small, the aspect ratio is high, and it has a needle-like interlocking structure. The grain boundary ferrite (GBF) structure is defined as an equiaxed structure with a large size and a close aspect ratio. It is noticeable that a large number of fine inclusions was dispersed in the weld metal after austenitization at 1400 °C for ten seconds and cooling at 1 °C/s. A large number of GBF and a very low proportion of AF were induced by these inclusions [18,19]. With the increasing cooling rate, AF was formed inside grains, whereas coarse GBF was formed at grain boundaries. The proportion of AF increased as the size of GBF decreased. When the cooling rate was 100 °C/s, no AF was formed in the weld metal (blocky equiaxed ferrite only existed).

Figure 3 presents the AF proportion in each sample with different cooling rates, the AF proportion in the weld metal gradually increased 20%, 45%, and 80% at 1 °C/s, 5 °C/s, and 10 °C/s, respectively. However, when the cooling rate reached 100 °C/s, the AF proportion was almost zero, indicating that the AF driving force by inclusions changed dramatically when the cooling rate changed.

### 3.2. Precipitation of MnS and Concentration Distribution of Mn

Figure 4 exhibits the microstructure of the weld metal at the cooling rate of 10 °C/s. These inclusions induced multiple AFs. The inclusions mainly consisted of O, Ti, Al, Mn, and S. Ti and Al existed in the core of these inclusions, whereas Mn and S were distributed in the same position on the outer layer (determined as MnS). The meaning of the scale bars in the EPMA maps represents the signal strength of the corresponding elements.

Figure 5 displays the microstructure of the weld metal at the cooling rate of 100 °C/s. Inclusion with a size of approximately 0.25 µm did not induce AF. Similar to Figure 4, the inclusions, in this case, were also composed of O, Ti, Al, Mn, and S, and Ti and Al existed in the core. However, the main difference is that the distribution of MnS in the outer layer was too small. Hence, the precipitation of MnS on the surface of inclusions and the formation of AF were suppressed when the cooling rate reached 100 °C/s.

Figure 6 presents the TEM morphology of inclusions in the weld metal at the cooling rate of 10 °C/s. Multiple AFs with an interlocking arrangement were formed on inclusions, which is consistent with the results shown in Figure 4. The concentration distribution of Mn around inclusions was measured by EDS. In order to improve measurement accuracy, the Mn concentration at a position was measured multiple times, and the average value was taken as the final Mn concentration (Figure 6b).

Figure 7 shows that when the cooling rate of the weld metal is less than 10 °C/s, there is a region with a decrease in Mn concentration of about 50 nm near the matrix of the inclusion, which is usually called the Mn-depleted zone (MDZ). At 10 °C/s, the minimum Mn concentration in the matrix near the inclusions in the weld metal is 1.58%. With decreasing cooling rate, the minimum Mn concentration near the inclusions gradually increases, and the Mn concentrations at 5 °C/s and 1 °C/s are 1.64% and 1.71%, respectively. When the weld metal’s cooling rate reaches 100 °C / s, no decrease in Mn concentration is observed around the inclusions, and no MDZ forms.

## 4. Discussion

### 4.1. Calculation Model for MDZ Formation

It needs to be clear that the type of inclusions in this work are determined to be (Ti-Al)O_x_ composite inclusions as the core, and MnS is wrapped around the oxide. Considering that this work mainly studies the MDZ mechanism, the focus of the author’s research is also mainly on MnS inclusions. Therefore, the model proposed in Figure 8 is only for MnS. The calculation model used to detect the change in Mn concentration around inclusions is presented in Figure 8. The concentration of Mn and S in the γ-phase changed linearly. The precipitation of MnS was first calculated, and the concentration changes of Mn and S were then determined. The specific characteristics of the calculation model are listed below [20].

(a)The concentration change in Mn and S in the γ-phase was approximately linear.(b)On the surface of inclusions, the precipitation of Mn and S met the solid solubility of MnS and the local equilibrium condition.(c)It was assumed the diameter of inclusions was 1 µm, the distance between inclusions was 10 µm, and the initial Mn and S concentrations were 1.8% and 0.006%, respectively.(d)The short-term growth and precipitation of MnS on the surface of inclusions satisfied Equations (1)–(5) [21].Log(C_Mn_ C_S_) = A/T + B(1)(C_Mnθ_ − C_Mn0_)r = Δr_Mn_(C_Mn0_ − C_Mn_)/2(2)(C_Sθ_ − C_S0_)r = Δr_S_(C_S0_ − C_S_)/2(3)dr/dt*(C_Mnθ_ − C_Mn_) = D_Mn_(C_Mn0_ − C_Mn_)/Δr_Mn_(4)dr/dt*(C_Sθ_ − C_S_) = D_S_(C_S0_ − C_S_)/Δr_S_(5)
where C_Mn_ and C_S_ are the interfacial concentrations of Mn and S around inclusions (mol/cm^3^), r is the radius of inclusions (MnS) (cm), Δr_Mn_ is the size of MDZ (cm), Δr_S_ is the size of the S-depleted zone (cm), and C_Mnθ_ and C_S0_ are the initial concentrations of Mn and S, respectively (mol/cm^3^). The values of C_Mn_, C_S_, Δr_Mn_, and Δr_S_ in the cooling process of the weld metal were calculated repeatedly.

The concentration distributions of Mn around inclusions at different cooling rates are presented in Figure 9. During cooling, the concentration of Mn near inclusions decreased due to the precipitation of MnS on the surface of inclusions, and MDZ was gradually formed. The interfacial concentration of Mn around inclusions in the weld metal was the lowest, and MDZ was the smallest when the cooling rate was 10 °C/s. However, as the cooling rate was reduced, the interfacial concentration of Mn gradually increased, and the range of MDZ was expanded. The interfacial concentration of Mn around inclusions did not decrease when the cooling rate reached 100 °C/s. These calculated results are consistent with the actual measurements, revealing the influence of cooling rates on the distribution of Mn in the weld metal.

### 4.2. Effects of Cooling Rates on the Concentration Distribution of Mn in MDZ

Ti and Al elements existed in the core of inclusions as they have strong thermal stability, whereas Mn and S elements existed in a solid-solution state during the heating process. As the temperature decreased, Mn and S atoms diffused around oxides to form MnS [22,23,24]. The concentration distribution of Mn around inclusions in the weld metal was characterized by TEM-EDS and the calculation model. The calculated results were found to be consistent with the TEM-EDS results, indicating that the calculation model could better reflect the change in Mn concentration in the weld metal during continuous cooling. The diffusion rate of Mn atoms was slower than that of S atoms; thus, MDZ was formed by the precipitation of MnS. MnS would not precipitate after a certain period due to the disappearance of S. Mn in the matrix also could diffuse to MDZ, causing the expanding of MDZ during cooling. When the cooling rate was less than 10 °C/s, the solubility of MnS decreased with the decreasing temperature, and MnS started to precipitate on the surface of inclusions, forming an Mn-depleted zone. MnS did not precipitate further after S completely precipitated as the same content of S element in weld metals; the amount of MnS precipitated on the surface of the inclusion is not significantly different, resulting in the same content of Mn element diffused from γ-phase to the surface of the inclusion, and the range of Mn-depleted zone formed in γ is consistent [24,25,26].

When MnS stopped precipitating during the cooling process, a concentration difference of Mn elements occurred between the γ matrix and MDZ. Mn atoms in the matrix slowly diffused to MDZ; thus, the interfacial concentration of Mn around inclusions increased, and the range of MDZ was expanded. When the cooling rate was moderate, Mn atoms required more time to diffuse and the diffusion effect was more significant. Therefore, at the cooling rate of 10 °C/s, the interfacial concentration of Mn around inclusions in the weld metal was the lowest, and the range of MDZ was the smallest. The diffusion of Mn atoms was impeded by faster cooling rates, hindering the precipitation of MnS on the surface of inclusions (Figure 5); thus, no MDZ was formed in the weld metal.

### 4.3. Effects of MDZ on AF Nucleation

Generally, the nucleation of AF is influenced by inclusion distribution, cooling rate, and AF nucleation energy [27,28]. In this study, the distribution of inclusions and the cooling rate during AF formation remained the same in all cases, only the energy for AF nucleation was different. The area ratio of AF changed significantly after the continuous cooling process. The area ratio of AF and the interfacial concentration of Mn around inclusions were strongly correlated, and it was found that the lower the interfacial concentration of Mn, the higher the AF proportion. No MDZ was formed in the weld metal at the cooling rate of 100 °C/s; thus, the proportion of AF was zero, indicating that MDZ had a noticeable influence on AF formation.

Under high heat input welding, GBF generally nucleate at grain boundaries. When the driving force for AF is achieved; AF can be formed inside grains as temperature decreases [29]. In this study, when the weld metal was cooled at a rate of 10 °C/s, the interfacial concentration of Mn around inclusions was the lowest, and the decrease in Mn concentration was the largest. Moreover, at the cooling rate of 10° C/s, the phase transition point of AF in the weld metal was the highest. When the weld metal was transformed from γ to α, equiaxed GBF at grain boundaries induced the phase transition of AF. The phase transition time of GBF became very short, and the proportion of GBF decreased; thus, AF was the predominant element in the weld metal.

As the cooling rate decreased, the interfacial concentration of Mn around inclusions gradually increased, the decrease in the interfacial concentration of Mn started to decrease, and the phase transition point of AF also decreased. When the phase transition occurred from γ to α, the transition time of equiaxed GBF was prolonged, and the phase transition occurred at grain boundaries. The phase transition occurred from grain boundaries to the grain interior until the temperature decreased to the phase transition point of AF [30].

The microstructure of the weld metal at the cooling rate of 1 °C/s is displayed in Figure 10. GBF in the γ-phase started to nucleate at grain boundaries when the weld metal was cooled to 642 °C (Figure 10a). As the temperature decreased, GBF gradually grew toward the grain interior, GBF near grain boundaries also started to nucleate. Inclusions inside grains do not induce AF, whereas inclusion 4 near the grain boundary lead to GBF transition (Figure 10b). When the temperature of the weld metal decreased to 616 °C, AF started to nucleate. Inclusion 1 inside the grain began to induce the nucleation of AF, followed by inclusions 2 and 3 (Figure 10c). AF induced by inclusion 1 continued to grow (Figure 10d), causing the AF nucleation to become restricted; therefore, the proportion of AF decreases. This is the main reason why the proportion of AF in the weld metal decreased, and the proportion of equiaxed ferrite increased continuously as the cooling rate decreased. Furthermore, at the cooling rate of 100 °C/s, no MDZ was observed in the weld metal, and no AF transformation occurred inside grains; thus, the proportion of AF was zero.

Figure 11 shows the microstructure characterization of the weld at a typical cooling rate (10 °C/s). It can be seen that acicular ferrite (AF) and grain boundary ferrite (GBF) are formed in the weld metal at a cooling rate of 10 °C/s. The acicular ferrite structure has an interlocking structure of high-angle grain boundaries, which can refine the grains and effectively deflect the crack propagation.

## 5. Conclusions

(1)According to the difference of cooling rate in austenitizing temperature stage, MnS with different precipitated amount can be constructed. The volume fraction of acicular ferrite is directly affected by the range of the Mn-Depleted zone surrounding the inclusions and the concentration distribution of Mn element.(2)When the cooling rate is fast (100 °C/s), the precipitation of MnS is the least, the Mn-depleted zone cannot form around the inclusions, and the inclusions cannot effectively induce the formation of acicular ferrite.(3)When the cooling rate is 10 °C/s, the range of the Mn-depleted zone around MnS is the smallest, the interface concentration of Mn element around inclusions is the lowest, and the volume fraction of acicular ferrite is the highest. When the cooling rate is 5 °C/s and 1 °C/s, the lower the cooling rate, the higher the interface concentration of Mn element, the larger the range of the Mn-depleted zone, and the lower the proportion of acicular ferrite in weld metal.

## Figures and Tables

**Figure 1 materials-15-08477-f001:**
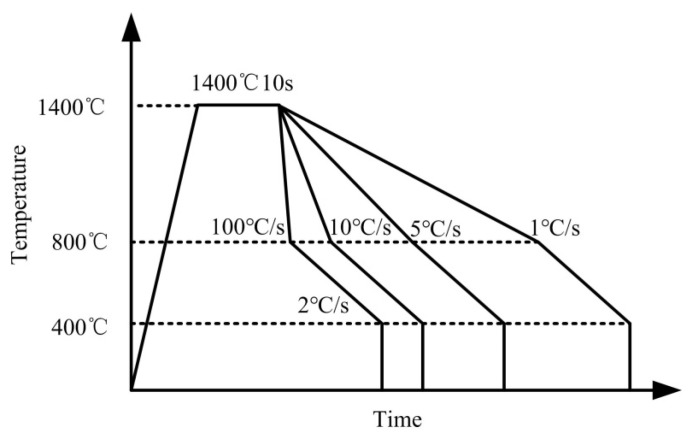
Schematic illustration of the heat treatment process of the weld metal.

**Figure 2 materials-15-08477-f002:**
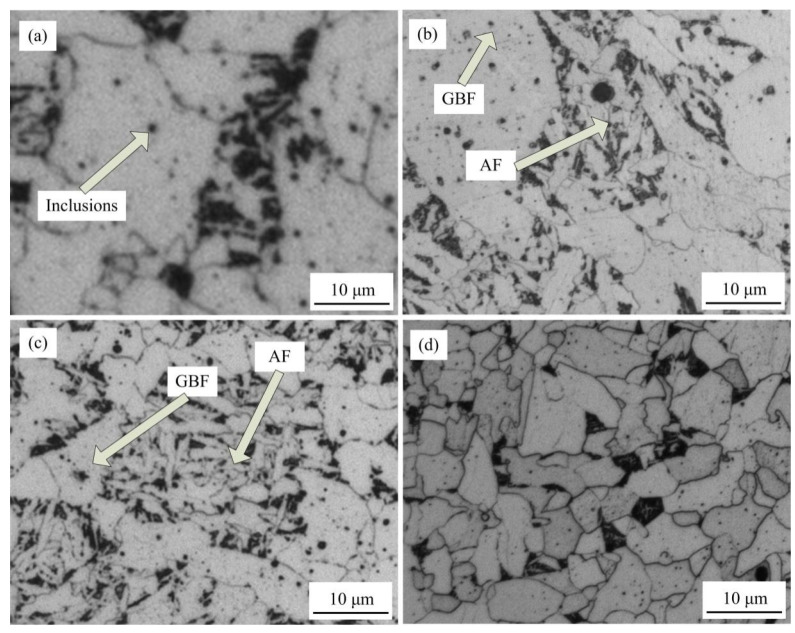
Microstructures of the weld metal after continuous cooling at different rates: (**a**) 1 °C/s, (**b**) 5 °C/s, (**c**) 10 °C/s, and (**d**) 100 °C/s.

**Figure 3 materials-15-08477-f003:**
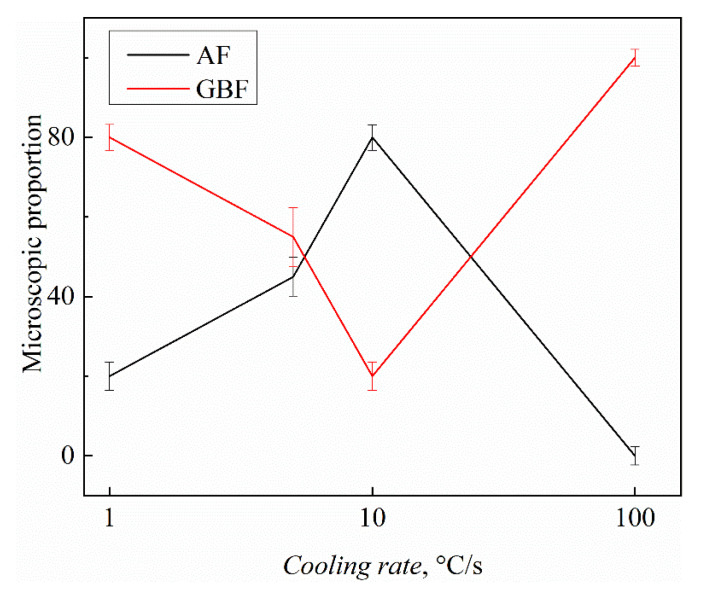
The microscopic proportion of AF and GBF in the weld metal at different cooling rates.

**Figure 4 materials-15-08477-f004:**
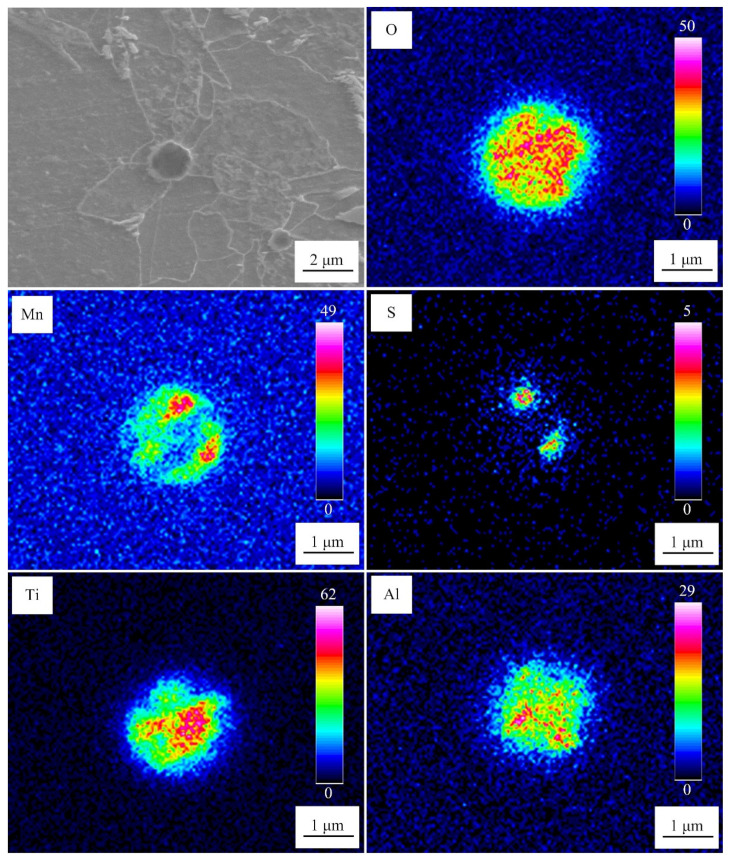
SEM-EDS mapping image of inclusions in the weld metal at the cooling rate of 10 °C/s.

**Figure 5 materials-15-08477-f005:**
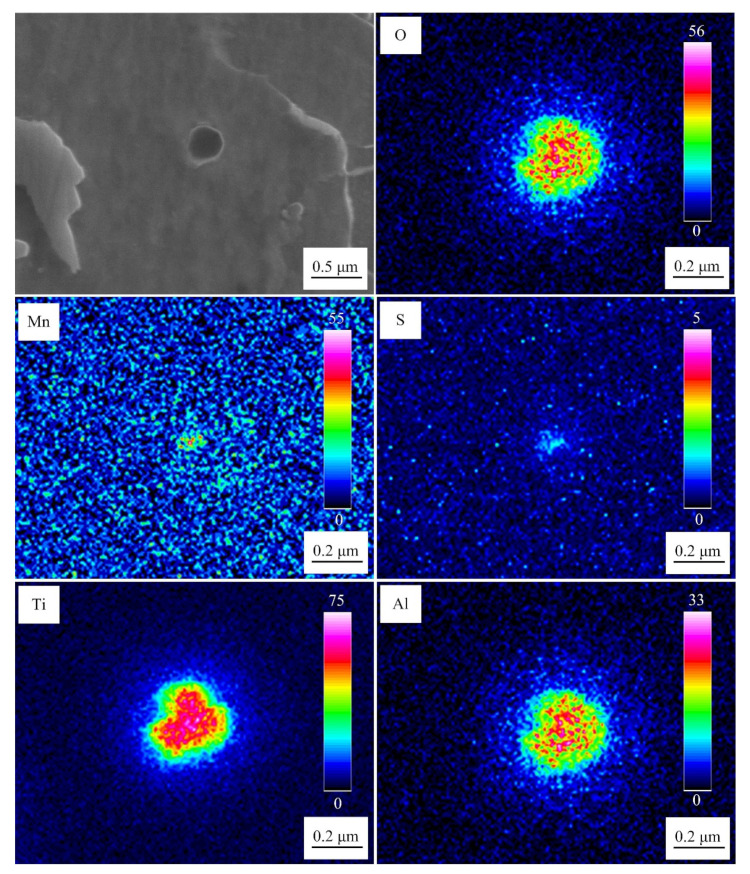
SEM-EDS mapping image of inclusions in the weld metal at the cooling rate of 100 °C/s.

**Figure 6 materials-15-08477-f006:**
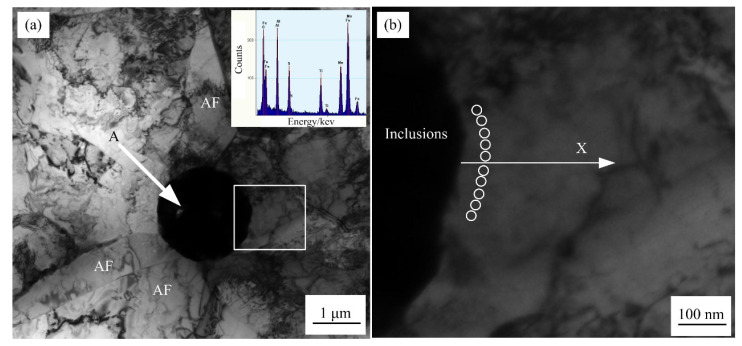
TEM morphology of inclusions in the weld metal at the cooling rate of 10 °C/s: (**a**) Original image with EDS and (**b**) locally magnified image.

**Figure 7 materials-15-08477-f007:**
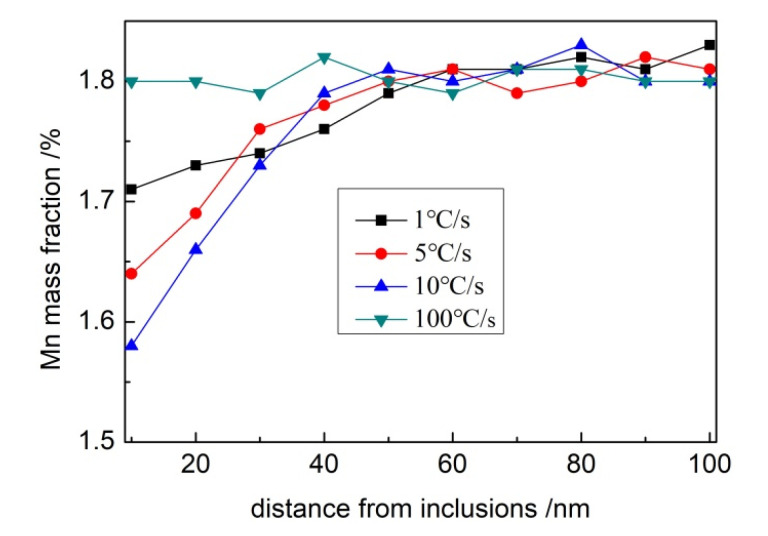
Concentration distribution of Mn around inclusions in the weld metal.

**Figure 8 materials-15-08477-f008:**
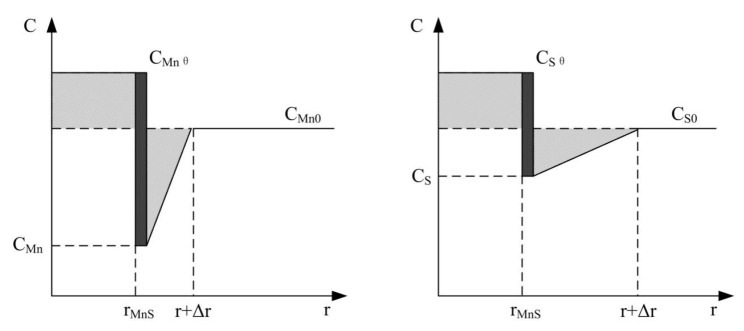
Calculation model of Mn and S concentrations in the γ-phase.

**Figure 9 materials-15-08477-f009:**
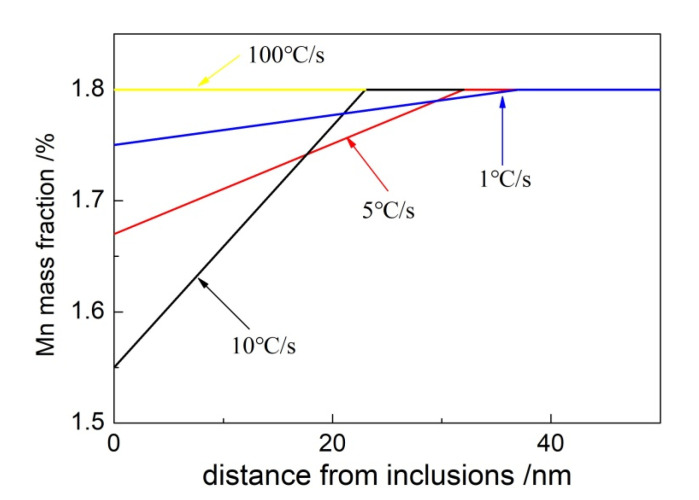
Calculation results of Mn concentration around inclusions.

**Figure 10 materials-15-08477-f010:**
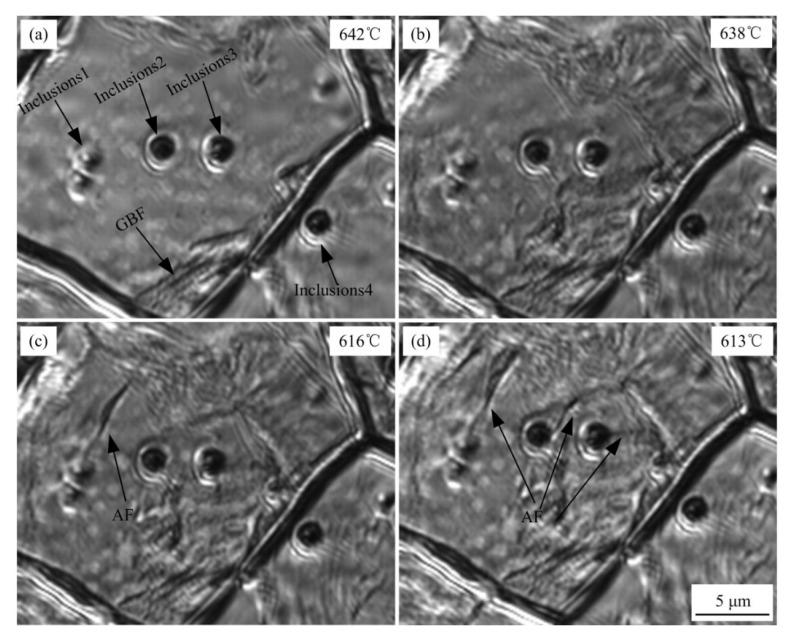
Microstructural evolution during γ → α phase transformation in the weld metal at the cooling rate of 1 °C/s. (**a**) cooled to 642 °C, (**b**) cooled to 638 °C, (**c**) cooled to 616 °C, (**d**) cooled to 613 °C.

**Figure 11 materials-15-08477-f011:**
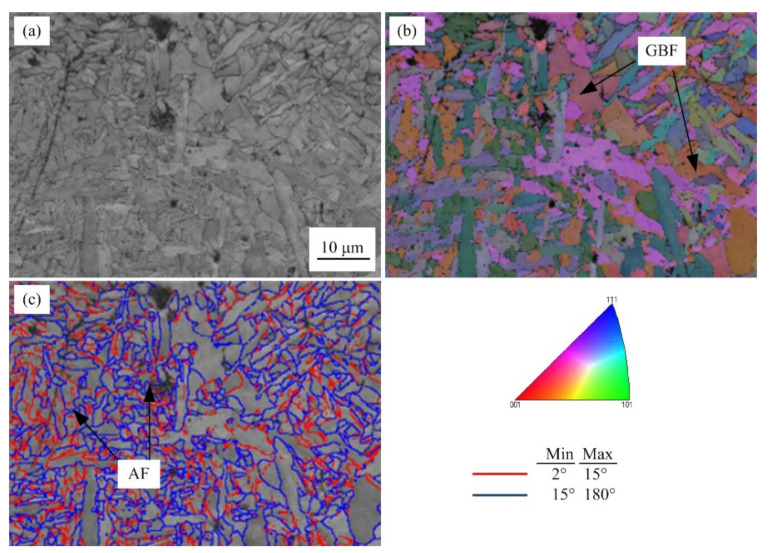
EBSD maps of weld metals at the cooling rate of 10 °C/s. (**a**) Image quality maps, (**b**) inverse pole figure maps, and (**c**) grain boundaries of misorientation angle >15° (blue) and 2–15° (red).

**Table 1 materials-15-08477-t001:** Chemical composition of the weld metal (mass (%)).

C	Si	Mn	Al	S	B	O	Ti	Fe
0.08	0.3	1.8	0.41	0.006	0.008	0.06	0.08	Bal.

## Data Availability

Not applicable.

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
