# Peer review of "Effects of Mn-Depleted Zone Formation on Acicular Ferrite Transformation in Weld Metals under High Heat Input Welding"

_materials, 2022, doi:10.3390/ma15238477_

Round 1

Reviewer 1 Report

Dear Author,

Please find below my suggestions for improvement, take it forward with interest.

1)acicular ferrite can form in reheated weld metals when the austenite grain size is relatively large- This concept is not promoted well 

2) What is the novelty of this work, this should be given clearly either in introduction or in abstract.

3) It is mentioned as high heat welding, mention what temperaturee is used in abstract itself.

4) Citation pattern used by author is not the journal pattern, please change and resubmit it

5) I feel authors used mass citation without real meaning, hence the article need to be returned for correcting this 

6) Introduction is very poor, complete revamping is suggested.

7) Grammar issue exists in many parts of the article.

8) Why samples were heated and cooled at same time, what is significance of it not clearly depicted in this work

9) Explain importance of fig 1 clearly, this is important for this work.

10) What is this "nitric acid alcohol" I have never heard of this type of alcohol.

11) Is fig 3 title clear and proper?

12) Conclusion is poorly formed, not all conclusions from discussions is clearly given

Reviewer 2 Report

The formation of AF is inherently related to cooling rate as well as composition. Authors contribute the formation of AF to MDZ of MnS is not correct. Especially for the size of MDS is so tiny in the TEM examination. There are many specific questions as listed below:

1.   What is the chemical composition of base metal? There is a dilution effect between filler metal and base metal. Actually dilution of base metal is related to weld metal. In table 1, Fe should be included as balance.

2. AF and GBF are not well distinguished.  How Fig. 3 is measured?

3. Fig. 4 is not acceptable. The scale bar is not correct. Please use quantitative mappings of WDS for all elements because their concentrations are quite different.

4. The concentration of S is as low as 0.006 wt%. Most inclusions in the weld metal are not MnS. The driving force of MDZ is not correct if the inclusion is mot MnS.

5. Most inclusions are oxides, NOT MnS. The proposed model in Fig.8 is not true for oxides. 

6. The formation of AF  in Fig.10 is not clear. The AF is not well identified. EBSD could be an appropriate tool in the analysis.

Reviewer 3 Report

The observations on this article are as follows:

1. The novelty of the work is not clear and there are lot of literature like this study. A comparison table should be included to confirm the originality of this work.

2. Detailed selected area diffraction (SAD) analysis in TEM should be included and using this perception phase of inclusions phenomena should be clearly expressed. 

3. How the acicular ferrite (AF) formation effects the Mn depleted zone and how this work can be differentiated with the work of Fujiyama, Naoto, and Genichi Shigesato. "Effects of Mn and Al on acicular ferrite formation in SAW weld metal." ISIJ International 61, no. 5 (2021): 1614-1622.. 

4. Acicular ferrite formation on grain refinement in the coarse-grained region of heat-affected zone-explain with some images. 

5. Mn near the depleted zone for various thermal cycles should be mentioned clearly.

 In summary, I am worried about the work as there are lot of similar types of works are already available. So I think the work is  rigorous  and some recent and different types of analysis in relation to application in industry clearly be explored. 

Round 2

Reviewer 2 Report

Most of my concerns are explained. Only a few questions left as below:

1. What is the solidus/liquidus of the weld metal? I need to make sure that weld metal at 1400 C is in liquid without solid phase.

2.  In Fig. 3, is it possible to include standard deviation in every data point?

3. In Figs. 4 and 5, authors need to explain what is the meaning of the scale bars? For example, is the maximum oxygen concentration in Fig. 4 is 50 at%? If not, please explain it in the text.

4. The citation of Equation 1 ~5 must be included in the text. In my opinion, the concentration of S was too low to form MnS, and I consider the concentration oS is the rate controlling factor in MnS formation.

Reviewer 3 Report

The article can be accepted in present form

Author Response

Thanks again for your helpful comments.